# Changing Characteristics of Tropical Extreme Precipitation–Cloud Regimes in Warmer Climates

William K. M. Lau [1,*], Kyu-Myong Kim [2], Bryce Harrop [3] and L. Ruby Leung [3]

1   Earth System Science Interdisciplinary Center, University of Maryland, College Park, MD 20740, USA
2   Climate and Radiation Laboratory, NASA/Goddard Space Flight Center, Greenbelt, MD 20771, USA; kyu-myong.kim@nasa.gov
3   Pacific Northwest National Laboratory, Atmospheric Sciences and Global Change Division, Richland, WA 99352, USA; bryce.harrop@pnnl.gov (B.H.); ruby.leung@pnnl.gov (L.R.L.)
*   Correspondence: wkmlau@umd.edu

**Abstract:** In this study, we investigated the changing characteristics of climatic scale (monthly) tropical extreme precipitation in warming climates using the Energy Exascale Earth System Model (E3SM). The results are from Atmospheric Model Intercomparison Project (AMIP)-type simulations driven by (a) a control experiment with the present-day sea surface temperature (SST) and $CO_2$ concentration, (b) P4K, the same as in (a) but with a uniform increase of 4K in the SST globally, and (c) the same as in (a), but with an imposed SST and $CO_2$ concentration from the outputs of the coupled E3SM forced by a 4xCO$_2$ concentration. We found that as the surface warmed under P4K and 4xCO$_2$, both convective and stratiform rain increased. Importantly, there was an increasing fractional contribution of stratiform rain as a function of the precipitation intensity, with the most extreme but rare events occurring preferentially over land more than the ocean, and more so under 4xCO$_2$ than P4K. Extreme precipitation was facilitated by increased precipitation efficiency, reflecting accelerated rates of recycling of precipitation cloud water (both liquid and ice phases) in regions with colder anvil cloud tops. Changes in the vertical profiles of clouds, condensation heating, and vertical motions indicate increasing precipitation–cloud–circulation organization from the control and P4K to 4xCO$_2$. The results suggest that large-scale ocean warming, that is, P4K, was the primary cause contributing to an organization structure resembling the well-known mesoscale convective system (MCS), with increased extreme precipitation on shorter (hourly to daily) time scales. Additional 4xCO$_2$ atmospheric radiative heating and dynamically consistent anomalous SST further amplified the MCS organization under P4K. Analyses of the surface moist static energy distribution show that increases in the surface moisture (temperature) under P4K and 4xCO$_2$ was the key driver leading to enhanced convective instability over tropical ocean (land). However, a fast and large increase in the land surface temperature and lack of available local moisture resulted in a strong reduction in the land surface relative humidity, reflecting severe drying and enhanced convective inhibition (CIN). It is argued that very extreme and rare "record-breaking" precipitation events found over land under P4K, and more so under 4xCO$_2$, are likely due to the delayed onset of deep convection, that is, the longer the suppression of deep convection by CIN, the more severe the extreme precipitation when it eventually occurs, due to the release of a large amount of stored surplus convective available potential energy in the lower troposphere during prolonged CIN.

**Keywords:** climate-scale extreme tropical precipitation; stratiform and convective precipitation; precipitation efficiency; meso-scale convective complex; surface warming vs. moistening; convective inhibition over land



## 1. Introduction

Recent reports of devastation resulting from record-breaking heavy precipitation around the world have provided strong indications that humanity is already experiencing

the disastrous effects of increased extreme precipitation, i.e., flash floods, soil erosion, land-slide, degradation of the eco-system, destruction of properties and loss of lives, attributable to anthropogenic greenhouse warming. Without a timely reduction in the emissions of greenhouse gases, the current trend in extreme precipitation will continue, and adverse impacts on the socio-economic system are likely to become worst [1]. Extreme precipitation in the tropics not only adversely affects the livelihood of more than 40% of the world population but is also a primary driver of global climate variability and change [2–5]. Hence, a better understanding of the physical processes underlying tropical extreme precipitation and its global impacts is paramount for the development and implementation of effective adaptation and mitigation strategies for global climate variability and change.

In the tropics and subtropics, climatologically strong surface heating and low-level moisture convergence lead to increased convective instability, enhancing heavy precipitation preferentially in regions with a warm surface temperature, i.e., the Inter-Tropical Convergence Zone (ITCZ), monsoon regions, and the maritime continent [6–9]. Changes in precipitation under global warming generally follow a geographic distribution pattern of "wet-gets-wetter" and "warmer-gets-wetter" [10–15]. A necessary condition for precipitation is the formation of clouds. Both precipitation and clouds, and their associated temporal and spatial distributions, are strong functions of atmospheric heating/cooling and moistening/drying processes, modulated by the surface temperature, heat and moisture fluxes, cloud microphysics, convection, and large-scale circulation [16–22]. Previous research on precipitation and clouds under climate variability and change have emphasized: (a) regional extreme precipitation events, cloud microphysics, and latent heating and forcing by a mesoscale convective system (MCS) [23–29], and (b) radiation heating feedback by various cloud types in determining global climate sensitivities [30–37]. While much knowledge has been gained and both approaches need to be continued in order to narrow down uncertainties, an emerging paradigm is that a deeper understanding of the myriad factors leading to extreme precipitation under climate change is predicated on a more comprehensive approach based on the broader context of interactions and enhanced by feedback processes involving cloud radiation, convection, and large-scale circulation [38–44].

Previous observational and climate modeling studies have shown that under global warming, the rate of increase in the top 0.1% of tropical daily precipitation has been estimated to be near 10% $K^{-1}$, significantly higher than those in the extratropics, which is limited by a thermodynamic rate of 6–7% $K^{-1}$, governed by the Clausius–Clapeyron relationship for atmospheric saturated moisture and temperature [2–5]. Models and observations have also shown that as Earth's surface and the atmosphere warm up under anthropogenic $CO_2$ radiative forcing, convection becomes more vigorous, and clouds grow faster, wider, and taller, producing more extreme precipitation [45]. An increasing number of recent studies [5,46–48] have shown that extreme precipitation events attributable to GHG warming tend to occur preferentially in tropical/subtropical regions with a strong and sustained organization of deep convection embedded in extended areas of high anvil clouds associated with long-lived strong mesoscale convective systems (MCS). Even though such long-lived MCS occur in less than 5% of the tropical precipitation events in preferred climatological wet regions, they account for more than 40% of the extreme precipitation amount [49]. This could mean that extreme precipitation, which occurs on hourly/daily time scales, could have organization signals on monthly and longer time scales over specific land or oceanic regions, and even over the entire tropics.

In spite of the increasing reports on devastating and destructive impacts on populated land regions, the scientific question of whether extreme cloud–precipitation organization is (a) fundamentally different and (b) more or less intense and/or frequent over land vs. ocean on climatic time scales remains uncertain. In this paper, we focus on addressing these questions and the scientific rationales underlying them based on general circulation model (GCM) simulations. However, because of the GCM's coarse resolution (>50–100km), MCS are not explicitly resolved and not well simulated in traditional GCM cloud–precipitation parameterizations. More recently, MCS-like features have been simulated and tracked in a

moderate resolution (50 km) GCM, with improved physical cloud–precipitation parameterization that includes organization features occurring across the scales [50]. In this study, we conducted AMIP (Atmospheric Model Intercomparison Project)-type [51] simulations using the Department of Energy's Exascale Energy Earth System Model (E3SM), which includes an improved unified parameterization of clouds and precipitation types, to examine its capability in simulating MCS-like features and contributing to extreme tropical precipitation on climatic time scales. See a further discussion on the E3SM model's physics in Section 2.

Specifically, we disentangled the effects of surface warming vs. atmospheric heating and moistening by increased $CO_2$ radiation forcing, leading to an occurrence of extreme precipitation–cloud regimes, with respect to changes in the stratiform vs. convective precipitation, precipitation efficiency, and thermodynamic vs. dynamical forcing over land and ocean. The organization of the paper is as follows. In Section 2, we describe the methodology, including the key physical parameterizations of the clouds and precipitation processes and the experimental design of the E3SM model experiments. In Section 3, we present the key results of the experiments. The conclusions and scope of continuing work are discussed in Section 4.

## 2. Model Description and Methodology

The U.S. Department of Energy (DOE)'s Energy Exascale Earth System Model Version 1 (E3SMv1) [52] was developed with the aim of addressing the grand challenge of actionable prediction of the Earth system's variability and changes to meet scientific and societal needs. The E3SMv1 is a fully coupled ocean–atmosphere–land–biosphere model, developed on the foundation of the Community Earth System Model version 1 (CESM1), but it includes adaptations and improvements to optimize the computational performance and science/application requirements of the DOE.

For clouds and precipitation, the E3SM atmospheric model (EAM) uses an improved version of Cloud Layers Unified by Binormals (CLUBB), which includes a third-order turbulence closure parameterization that unifies the treatment of boundary-layer clouds, shallow and deep convection, and cloud microphysics [53,54]. In the E3SM, improving the model of shallow cumulus clouds and stratocumulus clouds and precipitation was achieved by optimizing the scale dependence of the CLUBB parameterization for a diurnal cycle of precipitation over land [55]. Deep convective clouds and precipitation are based on the improved version of the Zhang and McFarlane (1995) [56] scheme, which included a recent update on the bulk parameterization of updraft processes (entrainment, detrainment, condensation, and precipitation) and downdraft processes (entrainment and evaporation of falling rain) from both liquid- and ice-phase precipitation [57]. Aerosol and cloud microphysics interactions in stratiform clouds are included in an updated version of the Modal Aerosol Module (MAM4) [58], which predicts the concentrations of major aerosol species (sulfate, black carbon, primary and secondary organic matter, mineral dust, and sea spray). The Morrison and Gettelman Version 2 [59] aerosol–cloud microphysics parameterization, coupled with CLUBB and MAM4, was used for the generation of shallow and stratiform clouds. The implementation of a convective gustiness adjustment to CLUBB significantly improved the simulation of stratiform and shallow clouds over the tropical ocean, where the climatological surface mean winds are weak [60]. Radiation–cloud–convection–circulation interaction (RC3I) processes in the EAM have also been significantly improved by better microphysics-based treatment of wet scavenging and re-suspension of evaporating precipitation, which affect the abundance and size of cloud condensation nuclei for liquid- and ice-phase precipitation, respectively [61]. In addition, this study used a variant of EAMv1 that adopted a consistent set of parameter adjustments, including sub-grid scale wind variance, resulting in better simulations of cloud properties [55].

Based on the Community Land Model (4.5) of CESM2, the land model (ELM) of the E3SM includes improvements in the representation of the water cycle processes of soil hydrology, river routing, coastal erosion, and biogeochemistry fluxes [52]. A new river

routing Model for Scale Adaptive River Transport (MOSART) was implemented, with particular emphases on human activities, including the management of water availability from river flow and the mitigation of floodplain inundation [62–64]. Two-way coupling between the MOSART and ELM was implemented to estimate the amount of water available from precipitation, river run-off, and storage in reservoirs for irrigation.

A key motivation for our analytical approach was to assess the degree to which the E3SM parameterization of fast and subgrid-scale cloud microphysical processes reflect the important contribution of mesoscale convective systems (MCS) to extreme cloud–precipitation organization on climatic (monthly and longer) time scales. Key features of MCS producing heavy precipitation over the ocean and land have been well documented [24,65]. During peak MCS development, a deep core with intense convective precipitation is coupled with extensive anvil clouds in the downwind regions, where stratiform precipitation dominates (Figure 1a). In the stratiform region, condensation heating associated with increased precipitation by active ice-phase microphysics (deposition, riming, and aggregation) causes a large-scale ascent in the upper troposphere above the freezing level (0 °C isotherm) near 500 hPa. At the same time, evaporative cooling by falling rain results in a large-scale mean descent in the lower troposphere. For tropical extreme precipitation events, the associated MCS life cycle may consist of multiple clusters of MCS complexes at various stages of development, starting with predominant convective precipitation and evolving to an increased contribution from stratiform precipitation. The results of cloud-resolving model simulations have shown that for non-MCS (100% convective) precipitation, the heating profile has a maximum near 500 hPa, while for "pure" stratiform precipitation, the heating profile shows a dipole structure with maximum heating (cooling) in the upper (lower) troposphere (Figure 1b). As a result, the degree of MCS development is reflected in the elevation of the level of maximum condensation heating, relative to that of convective (non-MCS) precipitation. However, it is important to note that the presence of an active convective core coupled with a substantial fraction of the stratiform (anvil cloud) region is essential for the development and maintenance of an MCS. A stratiform anvil cloud–precipitation regime decoupled from its convective core lacking in a sustained supply of ice-phase condensates from the convective core region represents a decaying MCS that readily dissipates and ceases to rain [24]. Hence, while the proportion of stratiform rain in a developing and active MCS is expected to increase with increasing extreme precipitation, it is unlikely to be close to 100% over the life cycle of the development of multiple organized MCS systems in extreme precipitation [65].

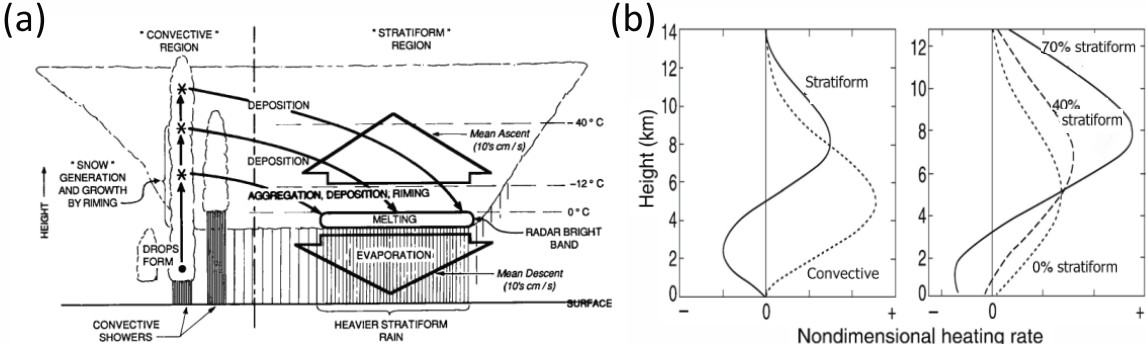

**Figure 1.** (**a**) Schematic showing organization of convective and stratiform cloud precipitation associated with a mesoscale convective system (adopted from Houze et al., 1989 [24]). Formation of water droplet and precipitation ice are denoted as ● and *, respectively. (**b**) Idealized vertical profiles of latent heating as a function of stratiform precipitation fraction from simulations of cloud-resolving models (adopted from Sui et al., 2020 [66]).

For model integration, the control experiment was represented by an equilibrium solution of an AMIP-type 10-year integration of the E3SM with a 100 km latitude–longitude resolution, and 72 layers with variable thickness in the vertical direction, with a top at 60 km,

under present-day sea surface temperature (SST), sea-ice conditions, and an atmospheric concentration of $CO_2$. To disentangle the effects of surface warming vs. $CO_2$ radiative forcing, equilibrium solutions based on two separate AMIP simulations identical to the control were conducted. First, the SST was increased by including an idealized plus-4K (P4K) anomaly uniformly across the globe. Second, an SST anomaly (SSTA) and $CO_2$ radiative forcing were imposed based on the climatology of the last 30-year simulation of an abrupt 4 times $CO_2$ (4x$CO_2$) experiment with the coupled ocean–atmosphere version of the E3SM, as part of the Coupled Model Intercomparison Project phase 6 (CMIP6) [67,68]. Changes in the tropical cloud precipitation characteristics for extreme precipitation were compared among the control, P4K, and 4x$CO_2$ simulations. Emphases were placed on a better understanding of the forcing and competing influences and feedback arising from surface warming vs. atmospheric heating and moistening processes. The realism of the model parameterization of the MCS and extreme precipitation, in terms of the changes in stratiform vs. convective precipitation, precipitation intensity, and large-scale circulation, was evaluated.

## 3. Results

### *3.1. Stratiform vs. Convective Precipitation*

Since climate models do not resolve clouds explicitly because of their coarse resolution, model precipitation is classified as "convective" if they are produced by the subgrid-scale parameterization of deep convection, and as "stratiform" precipitation if they are produced by condensation processes of the large-scale (LS) circulation represented by the cloud microphysics parameterization. In this paper, for convenience, we use the term LS rain fraction (LSRF) in the model as synonymous with stratiform rain fraction. The variations in the LSRF (Figure 2a) as a function of the monthly precipitation from January to December (J2D) show that there was an increasing contribution of LS rain as a function of the precipitation intensity (P) over the entire tropics in the control, P4K, and 4x$CO_2$ simulations, respectively. The LS rain fraction increased faster (steeper rise) in the order of control, P4K, and 4x$CO_2$. For very extreme precipitation (P > 30 mm/day), the LS rain fraction rose to above 50%, reaching a maximum of ~70%, for P > 40 mm/day under P4K and 4x$CO_2$. For convenience, the unit for P is omitted hereafter. In comparing the same plots but separated into ocean and land, it is clear that for P < 30, most of the increase in the LSRF came from the ocean (Figure 2b). This is not surprising because of the much larger area of ocean compared to land in providing precipitable water to the atmosphere. Over the ocean, increases in the LSRF as a function of P were more robust under 4x$CO_2$ compared to P4K, with the former showing a steady increase up to P < 30, and the latter showing a peak in the LSRF at P~20. As explained in later sections, the stronger signal under 4x$CO_2$ was likely due to stronger dynamical feedback under a physically consistent SSTA and additional $CO_2$-induced radiative forcing and response compared to the idealized uniform SSTA-only forcing under P4K. Most interestingly, in comparing the LSRF variation over land (Figure 2c) to those over land and ocean (Figure 2a) and over ocean only (Figure 2b), it is clear that almost all of the very extreme precipitation (P > 30), while occurring rarely, were found only over tropical land regions. Worth noting is that the LSRF seldom reached above 0.7 even over land, indicating that the generation and transport of ice-phase condensate by deep convection to the upper troposphere are essential in order for ice-phase microphysics generating stratiform rain to take place under the extended anvil clouds (see Figure 1a). Anvil clouds will dissipate quickly without the sustained generation of ice-phase condensation from the convective core [65,66].

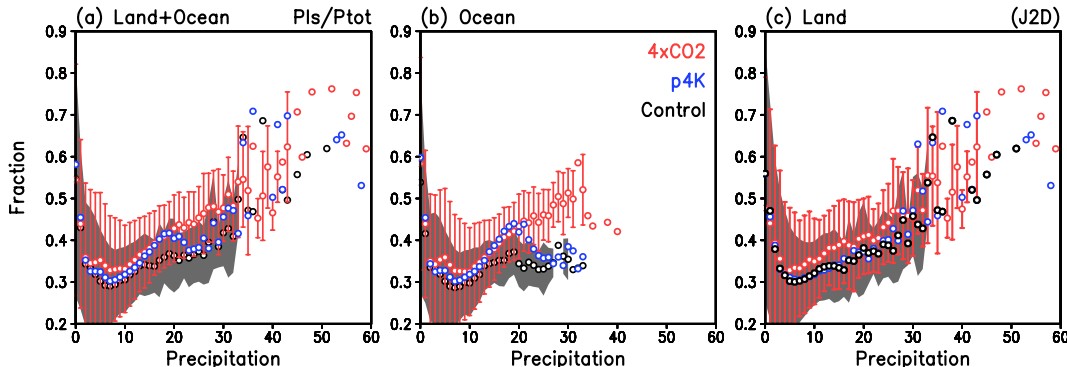

**Figure 2.** Stratiform rain fraction as a function of precipitation rate (mm day$^{-1}$) over (**a**) entire tropics, (**b**) ocean-only, and (**c**) land-only, for control (black), 4xCO$_2$ (red), and P4K (blue). Gray shading and red vertical bars represent a 1-s standard deviation for control, and 4xCO$_2$, respectively. Standard deviations for P4K are similar to 4xCO$_2$, and are not shown for clarity. J2D stands for monthly data taken from January to December.

A breakdown of the cumulative frequency of occurrence (FOC) of extreme monthly precipitation in terms of the total number of model grid points exceeding a given precipitation threshold (Table 1) shows that there was a rapid drop-off in the FOC with increasingly extreme precipitation. In the control climate, the FOCs of very extreme precipitation (P > 25–35) were indeed rarely (fewer than 1 in 1000) occurring preferentially over land, and rare or absent over the ocean. The FOCs of P > 30 increased by 3–5-fold under P4K and 4xCO$_2$ compared to the control and were stronger for the latter than the former. Analysis of the precipitation intensity threshold as a function of the top-percentile (PCT) rain rates showed similar signals, indicating more extreme heavy rain over land than the ocean (see Table S1). The preference for very extreme precipitation over land compared to the ocean appears to be an intrinsic property of the tropical ocean–land–atmosphere system, which was already present in the control, amplified under P4K, and even more so under 4xCO$_2$.

**Table 1.** Frequency of occurrence (FOC) measured in total number of model grid points over the entire tropics (30° S–30° N) as a function extreme monthly precipitation (P) exceeding a given threshold intensity, for control, P4K, and 4xCO$_2$ over land, ocean, and land + ocean, respectively. Quantities in bracket in first column show total number of grid points over the entire tropics (30° S–30° N). Unit of P is mm day$^{-1}$.

| | | P > 15 | P > 20 | P > 25 | P > 30 | P > 35 |
|---|---|---|---|---|---|---|
| Ocean (192984) | Control | 555 | 149 | 27 | 6 | 0 |
| | P4K | 2357 | 322 | 43 | 6 | 0 |
| | 4xCO$_2$ | 5555 | 1012 | 161 | 36 | 3 |
| Land (66216) | Control | 840 | 283 | 104 | 39 | 13 |
| | P4K | 1172 | 276 | 111 | 27 | 13 |
| | 4xCO$_2$ | 1739 | 581 | 244 | 117 | 51 |
| Ocean+ Land (259200) | Control | 1395 | 432 | 131 | 45 | 13 |
| | P4K | 3529 | 598 | 154 | 33 | 13 |
| | 4xCO$_2$ | 7294 | 1593 | 405 | 153 | 54 |

The spatial distributions of the frequency of occurrence (FOC) of extreme precipitation based on the rain rate for the top 1 percentile (PCT01) and top 5 percentile (PCT05) rainfall were computed. To facilitate comparison, the thresholds for the control for ocean + land were used to compute the FOC geographical distributions for P4K and 4xCO$_2$. The PCT01 (P > 13) rains (Figure 3a) occurred over limited areas within the climatological rainy regions of the Asian monsoon, the maritime continent/Pacific warm pool (SST > 302K), and the equatorial East Pacific ITCZ, with isolated signals over land regions in equatorial South

America and Africa. Under P4K (Figure 3b), the warm pool areas expanded substantially, covering much of the tropics. The PCT01 rain areas also expanded, but were still anchored to the climatological wet regions within the much warmer SST (SST > 304K). The increased PCT01 precipitation over the equatorial land region was more prominent compared to the control. Worth noting is that under P4K, except for the expansion of wetter areas, there were no fundamental changes in the spatial structure of tropical rainfall distribution compared to the control, suggesting a strong wet-getting-wetter (WeGW) scenario [10,11]. Under $4xCO_2$ (Figure 3c), the areal extent of the Pacific warm pool was further expanded compared to P4K, covering the entire tropical ocean (25° S–25° N). Over the aforementioned WeGW regions, PCT01 rain FOCs were further enhanced and expanded compared to P4K. Additionally, prominent centers of action for PCT01 precipitation were found over the equatorial Indian Ocean, and over the equatorial Atlantic Ocean under $4xCO_2$. that is, the expanded PCT01 rain areas exhibited not only WeGW but also a warmer-getting-wetter (WaGW) pattern [12]. Overall, the tropical SST was warmer by 1.85 K under $4xCO_2$ compared to P4K.

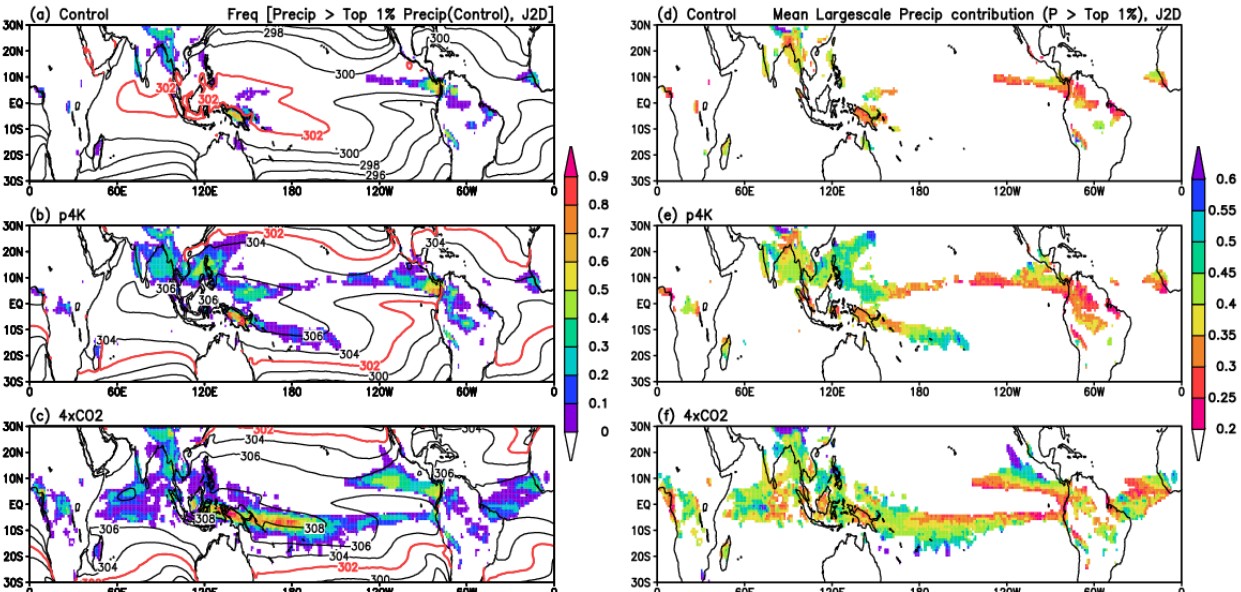

**Figure 3.** Spatial distribution of SST and frequency of occurrence (FOC) for top 1% precipitation (PCT01) in fractional units, based on monthly rainfall data from January through December (J2D), for (**a**) control, (**b**) P4K, and (**c**) $4xCO_2$, with warm pool SST (302K) outlined in red. Corresponding distributions for stratiform (large-scale) rain in fractional units are shown in (**d**–**f**), respectively.

Under the control climate, close matches between the areas of PCT01 rain (Figure 3a) and regions of an enhanced LSRF were discernable over the Asian monsoon land, the maritime continent, and the eastern Pacific ITCZ (Figure 3d). The sparse spatial extent of the PCT01 LSRF signals the rarity of such events in the control. Under P4K, the regions of enhanced FOC (Figure 3b) were well co-located with those with a large LSRF (>45–50%) over the Asian monsoon region, the maritime continent, the SPCZ, and the northern edge of the ITCZ over the eastern Pacific (Figure 3e). Under $4xCO_2$, the co-location of high FOCs of PCT01 rain (Figure 3c) with an increased LSRF (Figure 3f) could be seen over the aforementioned regions, as well as the land regions of equatorial Africa and the Amazon, consistent with the WeGW and WaGW patterns. Similar patterns of the FOC for PCT05 (P > 10) and an enhanced LS rain fraction were computed, indicating increasing contributions from LS (stratiform) rain types in more expansive regions of a high FOC compared to PCT01 (see Figure S1).

### 3.2. Precipitation Efficiency and MCS Organization

Recent model simulations and observations have shown that increased precipitation intensity is highly correlated with enhanced precipitation efficiency (PE), that is, an enhanced rate of microphysical auto-conversion of cloud water (liquid and ice phase), as the surface temperature rises [4,66,69–71], and it is a key driver of the large-scale circulation sustaining tropical heavy precipitation under global warming [72,73]. Here, we define the PE as the ratio of precipitation to the column integration of the total cloud water (TCW), including liquid and ice, as simulated by the microphysics parameterization of clouds and precipitation used in the E3SM (see discussion in Section 2).

$$PE = P/TCW \text{ (in units of s}^{-1}). \tag{1}$$

Physically, the inverse of PE ($\tau = PE^{-1}$) represents a characteristic residence time scale for the total condensed cloud water in an atmospheric column undergoing precipitation for a given precipitation rate. A high value of PE (low value of t) reflects vigorous water recycling within the atmosphere, converting cloud liquid and ice water into precipitation, while maintaining an abundant stock of the TCW in the atmosphere through enhanced surface moisture flux and low-level moisture convergence [66,74].

Figure 4a shows a nearly linear increase in the PE as a function of P over the entire tropics, with a faster rate (steeper gradient) of increase in the PE for extreme precipitation from the control to P4K to $4xCO_2$. The typical range of values of PE (0.02–0.2) is from $\tau = 50$ to 5 minutes, that is, there is a 10-fold reduction in the residence time scale of the TCW in the atmosphere, from light to the most extreme precipitation in the tropics. These values of $\tau$ can be considered a crude estimate of increasingly fast cloud–water–precipitation recycling time scales in MCS-like organization systems, contributing to the extreme precipitation in the E3SM model. Compared to the ocean-only plot (Figure 4b), it can be seen that most of the PE increase for P < 30 represents contributions mainly from oceanic precipitation, with a faster increase in the order of control, P4K, and $4xCO_2$. However, very extreme precipitation (P > 30) with high PE (PE > 0.1) was not found over the ocean. In contrast, the rate of increase in the PE as a function of P (Figure 4c) was faster over land than over the ocean for all precipitation rates. For extreme precipitation (P > 30) over land, the rate of increase in the PE was clearly accelerated compared to lower rain rates (Figure 4c). Comparing Figure 4a–c, it can be seen that almost all of the very extreme tropical precipitation (P > 30) and high PE (>0.1) events came from the land.

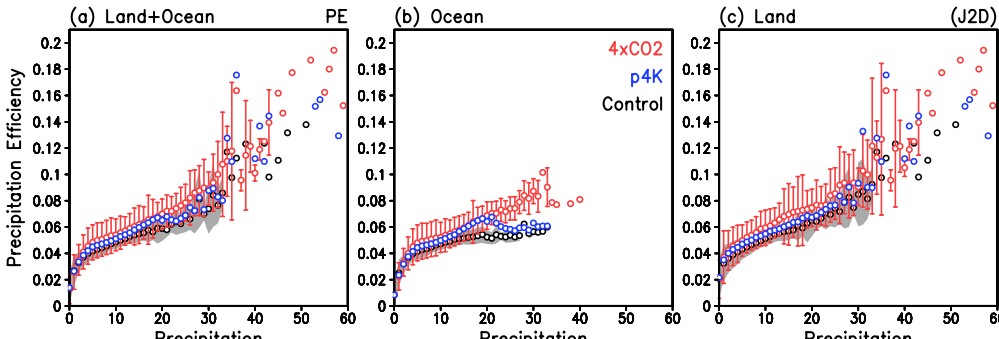

**Figure 4.** Precipitation efficiency (minute$^{-1}$) as a function of the precipitation rate (mm day$^{-1}$) for January through December, for (**a**) land + ocean, (**b**) ocean-only, and (**c**) land only, based on monthly data from January through December (J2D). Gray shading and red vertical bars represent 1 s standard deviation for control and $4xCO_2$, respectively. Standard deviations for P4K are similar to $4xCO_2$, but are not shown for clarity.

The geographic distributions of the PE of PCT01 rainfall for the control, P4K, and $4xCO_2$ (Figure 5a–c) show strong similarities to the pattern of outgoing longwave radiation (OLR), indicating an abundance of cold anvil clouds with low OLR (<190 Wm$^{-2}$) in regions

with enhanced PE (Figure 5d–f). Under P4K and $4xCO_2$, more so in the latter than the former, higher PE with lower OLR (more elevated clouds with colder tops) were found over the Asian monsoon, maritime continent, and equatorial Africa and South America regions. In contrast, over the open oceans of the Pacific ITCZ, the tropical western Pacific, and the South Pacific Convergence Zone (SPCZ), extreme precipitation was derived mostly from increased PE in regions with OLR $>215$ $Wm^{-2}$, consistent with an increased abundance of warm rain as a key signal of climate warming [69,75]. For moderately extreme precipitation (PCT05), the areal extent of high PE and low OLR increased substantially in conjunction with the expansion of the tropical SST warm pool (see Figure 3). Overall, the PE and OLR distributions for PCT01 and PCT05 exhibited the WeGW pattern under the control and P4K, and the WeGW + WaGW under $4xCO_2$, similar to those for the LSRF (see discussion about Figure 3).

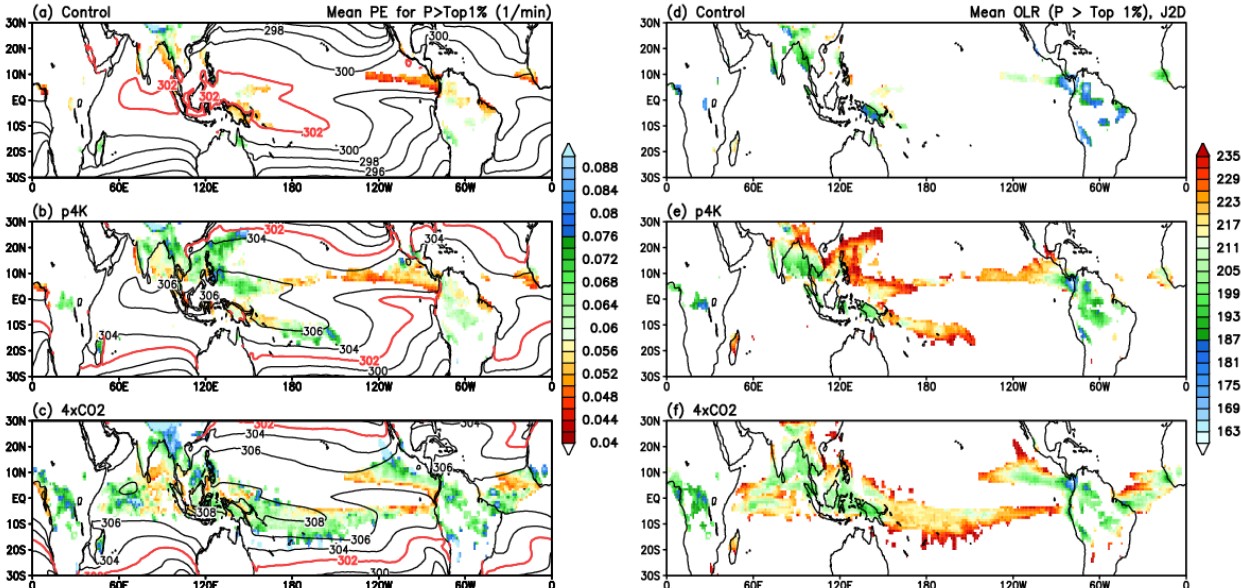

**Figure 5.** Spatial distribution of SST and precipitation efficiency (PE) for top 1% precipitation (PCT01) in units of minute$^{-1}$, based on monthly rainfall data from January to December (J2D) for (**a**) control, (**b**) P4K, and (**c**) $4xCO_2$, with respective warm pool SST outlined in red. Corresponding distributions for outgoing longwave radiation (OLR) in units of Watt m$^{-2}$ are shown in (**d**), (**e**), and (**f**) respectively.

Next, we explored the capability of the E3SM in simulating MCS-like extreme precipitation organization, with regard to an increased contribution from stratiform (anvil) rain, enhanced PE in the production by freezing, and the removal of cloud ice by melting and precipitation fallout. Specifically, we computed composite change patterns of P4K and $4XCO_2$ relative to the control and that of $4xCO_2$ relative to P4K in the vertical profiles of key MCS quantities, i.e., cloud ice concentration, condensation heating, and large-scale vertical velocity, as a function of the precipitation intensity of the entire tropics, separately for land and ocean. Over the ocean, the level of maximum cloud ice can be seen to shift upward relative to the control as precipitation increases under P4K (Figure 6a), starting at P~10 and continuing up to P > 25–30. The negative (positive) values of cloud ice signals accelerated the removal (accretion) of cloud ice below (above) 300 hPa by enhanced precipitation (condensation) relative to the control. Given the co-location of the regions of enhanced precipitation (PCT01) and the increased LSRF (Figure 3), as well as the increased PE and low OLR values (Figure 5), the cloud ice features are consistent with the enhanced model of MCS-like organization compared to the control and analogous to those shown in Figure 1a. Under $4xCO_2$ (Figure 6b), the cloud ice anomaly pattern is similar to that under P4K, indicating the primary importance of ocean warming in initiating the MCS organization. However, the MCS structure appears to be more robust under $4xCO_2$ compared

to under P4K. The stronger organized MCS development under $4xCO_2$ can also be seen in the difference plot of $4xCO_2$-minus-P4K (Figure 6c), indicating a stronger removal of cloud ice by precipitation near 400–250 hPa, and increased melting due to the warming of the middle and lower troposphere, coupled with enhanced cloud ice formation near 250–150 hPa. These likely reflect the effect of increased $CO_2$ radiative heating in the lower troposphere, enhancing convective instability in the upper troposphere [76]. Over land (Figure 6d,e), the changes in the cloud ice in the upper troposphere reflecting the increasing MCS organization under P4K and $4xCO_2$ are similar to those in the ocean, as is evident by the strong removal of cloud ice near 500–350 hPa and the accumulation of cloud ice above (250–150 hPa) associated with anvil cloud development. Under P4K and $4xCO_2$ (Figure 6d,e), the MCS organization over land shows less cloud ice loading (solid contours) but a more vertically confined region of negative anomalies, indicating stronger cloud ice removal compared to over the ocean. However, very extreme precipitation $P \geq 30$–35 occurred only over land in $4xCO_2$, but not over the ocean (solid contours in Figure 6b,e). The additional radiative heating effect due $4xCO_2$ further enhanced the MCS precipitation organization over land compared to P4K (Figure 6f).

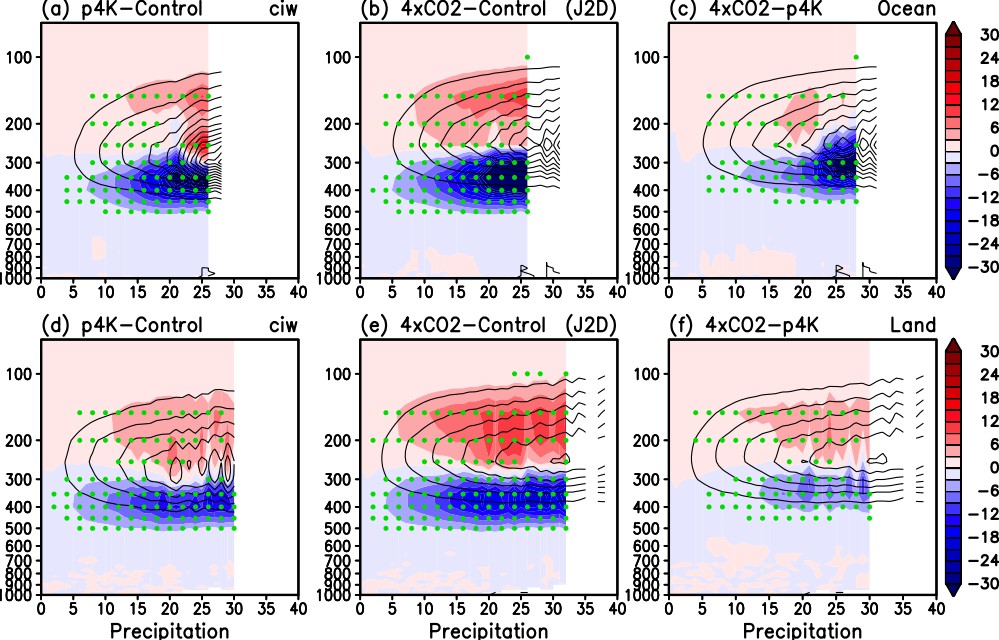

**Figure 6.** Vertical profiles of anomalous cloud ice contents ($10^{-6}$ kg/kg, ice mass per kilogram of air mass) as a function of precipitation intensity (mm day$^{-1}$) over ocean for (**a**) P4K-minus-control, (**b**) $4xCO_2$-minus-control, and (**c**) $4xCO_2$-minus-P4K, from January to December (J2D). Panels (**d**), (**e**), and (**f**) are the same as (**a**), (**b**), and (**c**), respectively, but over land. Contours show the mean profiles of condensation heating for the minuend (first term of the subtraction) indicated in the respective subpanel labels. Regions with statistical significance exceeding 95% confidence are highlighted by green dots.

Over the ocean, the condensation heating profiles as a function of P for P4K and $4xCO_2$ (Figure 7a,b) reveal an essential feature of MCS organization, that is, the elevation of the level of condensation heating is characterized by positive (negative) anomalies above (below) 300 hPa as the precipitation intensifies (cf. Figure 1b). This is consistent with the increase in the LSRF (see Figure 2) and PE (see Figure 4), as discussed previously. Strong cooling found near the freezing level at 500 hPa and regions slightly above signals enhanced melting and evaporation of falling rain. The MCS organization appears to be stronger under $4xCO_2$ relative to P4K, with more condensation heating above (below) 250hPa (Figure 7c). Over land, the condensation heating profiles (Figure 7d,e) exhibit similar features to their ocean counterparts, but with more robust MCS-like features, that

is, elevated condensation heating, strong cooling at the mid-troposphere freezing level and regions below (Figure 7d,e), and a stronger response in 4xCO$_2$ compared to P4K (Figure 7f) due to the additional radiative heating of the atmospheric CO$_2$.

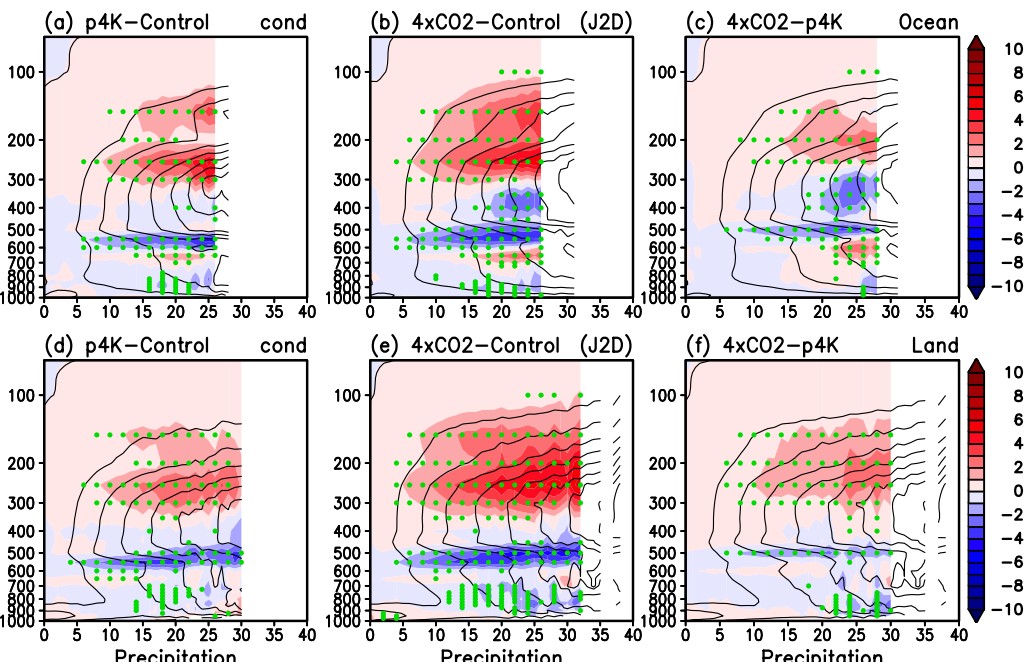

**Figure 7.** Vertical profiles of anomalous condensational heating (K day$^{-1}$) as a function of precipitation intensity (mm day$^{-1}$) over ocean for (**a**) P4K-minus-control, (**b**) 4xCO$_2$-minus-control, and (**c**) 4xCO$_2$-minus-P4K, from January to December (J2D). Panels (**d**), (**e**), and (**f**) are the same as (**a**), (**b**), and (**c**), respectively, but over land. Contours show the profiles of condensation of the minuend (first term of the subtraction) indicated in the respective subpanel labels. Regions with statistical significance exceeding 95% confidence are highlighted by green dots.

For the large-scale vertical velocity over the ocean under P4K and 4xCO$_2$ (Figure 8a,b), increased upward (downward) motions in the upper (middle-and-lower) troposphere are evident and consistent with the condensation heating (cooling) changes (see Figure 7). The decrease in the upward vertical motion in the mid-troposphere is indicative of the MCS organization, pertaining to an increased melting of cloud ice at the distinctive freezing level near 500 hPa and increased downdraft associated with evaporative cooling in the regions of falling rain (cf. Figure 1a). Again, the effects are stronger under 4xCO$_2$ compared to P4K (Figure 8c). Over land (Figure 8d,e), the changes in the large-scale vertical motions are similar to those in the ocean, except they appear more muted under both P4K and 4xCO$_2$, with the latter only slightly stronger than the former (Figure 8f). The stronger MCS-like signals over the ocean, especially the strong, distinctive cooling at the freezing level compared to over land, reflect the direct effects of stronger forcing over the ocean from the SSTA, as well as positive feedback from changes in the large-scale circulation. Importantly, the anomalous large-scale vertical motions over land shown here are likely attributable to not only the MCS organization but also changes in the large-scale Walker Circulation, driven by an east–west SST gradient and the land–sea thermal contrast, further modulating changes in the MCS convective updraft over land [77,78].

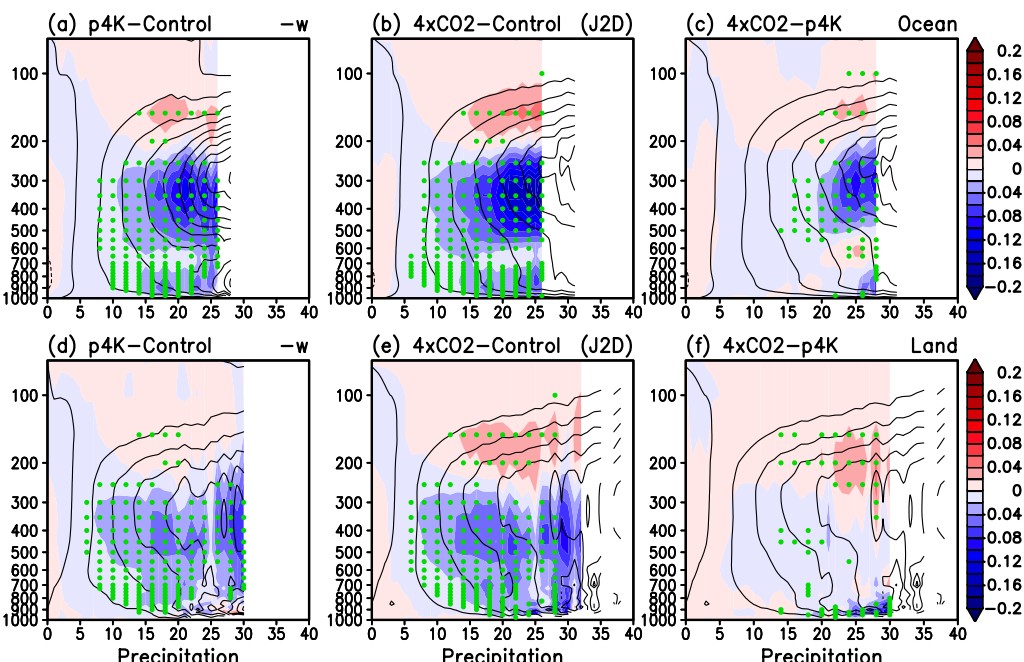

**Figure 8.** Vertical profiles of anomalous upward vertical velocity (Pa/s) as a function of precipitation intensity (mm day$^{-1}$) over ocean for (**a**) P4K-minus-control, (**b**) 4xCO$_2$-minus-control, and (**c**) 4xCO$_2$-minus-P4K, from January to December (J2D). Panels (**d**), (**e**), and (**f**) are the same as (**a**), (**b**), and (**c**), respectively, but over land. Contours show the profiles of condensation of the minuend (first term of the subtraction) shown in the respective subpanel labels. Regions with statistical significance exceeding 95% confidence are highlighted by green dots.

### 3.3. Convective Inhibition (CIN) and Extreme Precipitation

In this subsection, we explore further the fundamental reason why very extreme but rare (record-breaking) precipitation tends to occur over land rather than the ocean. Under GHG warming, the convective available potential energy (CAPE) is expected to increase due to the relative fast rate (~7% K$^{-1}$) of increase in the atmospheric saturated moisture with higher temperature. However, convective inhibition (CIN), that is, near-surface negative buoyancy, is known to be enhanced under global warming over land, resulting in increased drying (sub-saturation) of the near-surface air due to a lack of moisture supply relative to the fast land warming [79,80]. CIN drying is reflected in reduced low-level relative humidity, a higher lifting condensation level (LCL), and an elevated level of free convection (LFC), inhibiting deep convection [81].

To illustrate the effect of CIN under P4K and 4xCO$_2$ and its relationship with extreme precipitation, an analysis of the surface moist energy budget follows. The convective instability of the atmosphere is controlled by the vertical gradient of the moist static energy (MSE), with

$$MSE = C_pT + Lq + gz, \qquad (2)$$

where $C_p$ is the thermal capacity at a constant pressure, T is the surface air temperature, L is the latent heat of condensation, q is the specific humidity, g is the gravitation constant, and z is the geopotential height. A negative MSE vertical gradient (high—below, low—above) implies convective instability and vice versa for stability. For CIN, we focused on the first two terms (1) near the surface, that is, the lowest model level, where gz is negligibly small.

Under P4K (Figure 9a), the near-surface $C_pT$ anomalies (relative to the control) increased nearly uniformly (~4–5 kJ/kg) over the entire tropical ocean, following closely that of the imposed idealized 4K uniform SST warming. The $C_pT$ increase over land was stronger (~5–7kJ/Kg) compared to over the ocean because land has a lower thermal capacity than water. As a result, the land temperature rises faster and higher than that of the ocean with the same amount of heat input. In addition, the lack of land moisture sources

results in less evaporative cooling. Hence, under global warming, the land temperature has to rise much higher compared to the ocean temperature to enhance outgoing longwave radiative cooling, which is needed to balance the land heating from the $CO_2$ greenhouse effect and increased downward solar radiation from reduced clouds due to drying. The surface Lq (Figure 9b) followed a similar change pattern to $C_pT$ and was clearly the dominant forcing (~8–15kJ/kg), stronger than that of $C_pT$ by 2–3 times. This is because of the well-known exponential increase in the atmospheric saturated moisture content as a function of temperature, governed by the Clausius–Clapeyron relationship. Over the ocean, due to the readily available moisture from below, the near-surface relative humidity (RH) remained close to the saturation values. As a result, the anomalous relative humidity under P4K vs. the control was positive but small (<2–4%) over the ocean (Figure 9c). However, over land, because of the larger increase in the $C_pT$, the additional moisture required to reach saturation far exceeded that which could be derived from local moisture sources. As a result, there was a distinctive reduction in the RH (~3–6 %), indicating drying over land under P4K relative to the control (Figure 9c) and signaling increased CIN [81]. However, as the land temperature rises and CIN increases under P4K, the triggering of convection induced by mesoscale convergence, episodic outflow from land–sea breeze, and forced lifting from surface inhomogeneity and/or orographic may lead to an explosive growth of convection, releasing a large amount of stored CAPE during CIN [81]. The delayed onset of deep convection due to increased CIN could facilitate the occurrence of very extreme but rare precipitation in a warming climate, specifically over land. The stronger and longer-lasting the CIN, the more CAPE builds up in the lower troposphere, and the more extreme the precipitation when it eventually occurs, releasing a large amount of built-up CAPE during CIN [82–85].

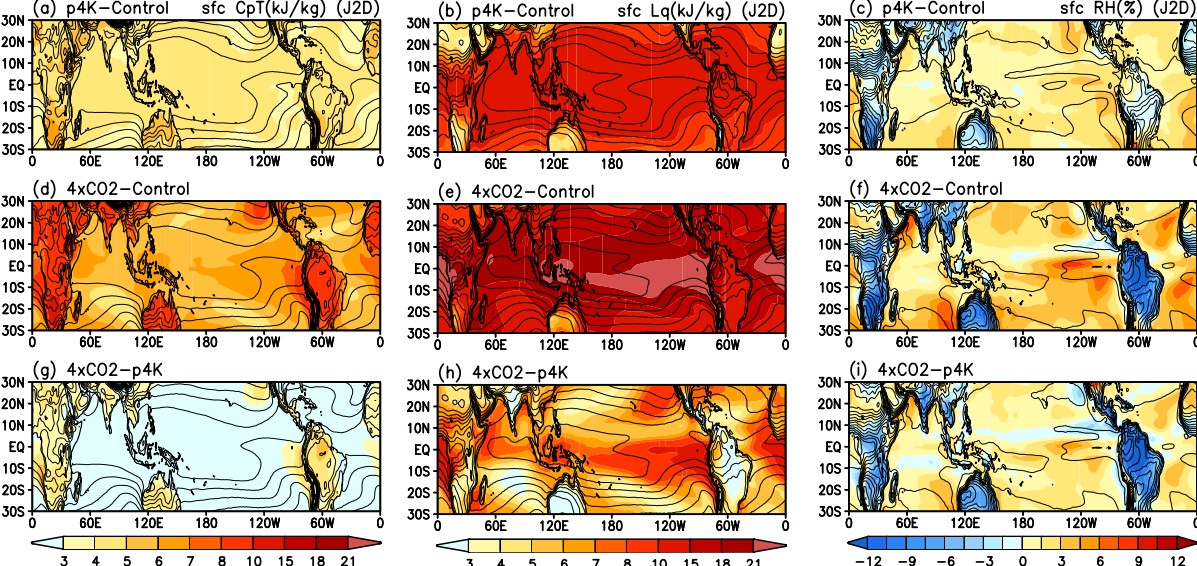

**Figure 9.** Spatial distribution of surface anomalous (**a**) $C_pT$ (kJ/kg), (**b**) Lq (kJ/kg), and (**c**) relative humidity (%) for January to December (J2D) under P4K. Panels (**d**), (**e**), and (**f**) are the same as in (**a**), (**b**), and (**c**), respectively, but under 4xCO2. Panels (**g–i**) are the same as (**a–c**) but for 4xCO2-minus-P4K.

Under $4xCO_2$ (Figure 9d), the near-surface anomalous $C_pT$ warming over land was much larger (>8–15 kJ/kg) than over the ocean (~5–7 kJ/kg). Again, the increase in the surface Lq (Figure 9e) over the tropical oceans followed the corresponding increase in the SST, consistent with $4xCO_2$ forcing and indicating enhanced warming and moistening of the surface air over the tropical ocean, following the Clausius–Clapeyron relationship. The differential magnitude of the anomalous Lq and $C_pT$ resulted in a large contrast in the relative humidity (RH) between the ocean and land, that is, increased RH over the tropical ocean, and decreased RH over land (Figure 9f), further enhancing the land–sea contrast, as

noted in P4K (Figure 9c). Judging from the 4xCO$_2$-minus-P4K pattern in C$_p$T (Figure 9g), it can be seen from the near uniform and small values (<3 kJ/kg) over the ocean that the P4K SST warming was a reasonable analog of the SST surface thermal forcing under 4xCO$_2$. However, the surface moisture forcing Lq reveals more regional features with higher values (>6–10 kJ/kg) over oceanic regions near the equator and subtropical monsoon regions adjacent to land in 4xCO$_2$ compared to P4K (Figure 9h). Clearly, enhanced atmospheric warming by 4xCO$_2$ further exacerbated the surface RH reduction over land compared to P4K (Figure 9i). This is likely due to enhanced land–atmosphere feedback arising from 4xCO$_2$, radiative forcing of the atmosphere and facilitated by cloud–convection–circulation interactions [19] under dynamically consistent SSTA forcing, increasing CIN, and the occurrence of very extreme but rare precipitation already operative under the control, but enhanced by P4K, and further amplified under 4xCO$_2$.

## 4. Concluding Remarks

Based on the AMIP-type model simulations using the Exascale Energy Earth System Model (E3SM), we investigated the changing characteristics of climate-scale (monthly) tropical extreme precipitation in a warming climate. Three ten-year-long AMIP-type model simulations were carried out: (1) a control, with the present-day SST, and CO$_2$ atmospheric concentration, (2) P4K, the same as the control but with a forced idealized uniform 4K increase in the SST globally, and (3) 4xCO$_2$, the same as the control but with SSTA derived from coupled model simulations under a four-times-higher atmospheric CO$_2$ concentration, including the corresponding 4xCO$_2$ radiative heating of the atmosphere. The key results of this study include the following:

- In a warming tropical climate, while both convective and stratiform rain increase, there is an increasing contribution from the stratiform rain fraction to extreme precipitation, with the most extreme but rare precipitation occurring preferentially over land compared to the ocean. However, the stratiform rain fraction approaches an upper limit of approximately 0.7, indicating that a deep convection core is essential to provide ice-phase condensate for stratiform rain even for the most extreme precipitation.

- The distributions of extreme precipitation (top 1% and 5%) generally follow the paradigms of wet-getting-wetter (WeGW) under the control and P4K, but both show WeGE and warmer-getting-wetter (WaGW) within an expanded tropical SST warm pool, and regional SST warming under 4xCO$_2$.

- Extreme precipitation is facilitated by increased precipitation efficiency (PE), reflecting an accelerated rate of recycling of precipitation and total cloud water (both liquid and ice phases) in regions of strongly reduced outgoing longwave radiation (<190Wm$^{-2}$), associated with colder (higher) anvil cloud tops.

- The increase in PE associated with the extreme precipitation under P4K and 4xCO$_2$ is reflected in a more MCS-like organization structure over land and ocean compared to the control, including (a) increased ice-phase upper-level clouds, (b) an elevated level of condensation heating in the upper troposphere and strong cooling from the enhanced melting of ice near the freezing level and altitudes below from the evaporation of falling rain, and (c) an increased ascent (descent) of large-scale vertical motion in the upper (lower) troposphere.

- Analysis of the surface moist static energy distribution revealed that moisture forcing (Lq) from an increased higher SST is the primary driver of extreme precipitation over the ocean, in accordance with the Clausius–Clapeyron relationship. However, surface temperature forcing (C$_p$T) is more important over land, as reflected in the much higher land surface temperature due to the smaller heat capacity of land and a lack of moisture sources from land.

- The high surface temperature over land leads to enhanced convective inhibition (CIN), that is, the drying of the land surface, reflected in reduced relative humidity of the near-surface air over land under P4K and 4xCO$_2$, more so in the latter than the former. We argue that the very extreme but rare precipitation over land is likely due

to increased CIN, delaying the triggering of deep convection, while building up the convective available energy in the lower atmosphere associated with the warming climate. When deep convection is triggered eventually through moisture advection from episodic small-scale atmospheric eddy processes associated with land–sea breeze, thunderstorms, and orography, an explosive growth of MCS-like organization occurs preferentially over land, releasing extra amounts of convective available potential energy (CAPE) stored during CIN, and resulting in very extreme "record-breaking" precipitation over land, as global climate warming continues unabated.

The similarities in MCS extreme precipitation development over ocean and land and between $4xCO_2$ and P4K underscore the importance of SST warming as the primary forcing in the development of MCS-like organization, leading to extreme precipitation. However, non-uniform SSTA based on ensemble coupled models together with dynamically consistent $CO_2$ radiative forcing of the atmosphere is needed to produce stronger and presumably more realistic regional characteristics of extreme precipitation in the warming climate of a future world through dynamical adjustments and feedback processes in the coupled atmosphere–ocean–land system. For a better understanding of the effects of CIN in staging very extreme "record-breaking" regional precipitation events over land, intrinsic land–atmosphere feedback processes and impacts by concomitant changes in the tropical large-scale circulation, land–sea contrast, and under P4K and $4xCO_2$, comparisons with CMIP6 model outputs, and multiple sources of precipitation and cloud observations are being investigated in our ongoing research.

Finally, we note that high-resolution MCS resolving meso-scale (10–20 km) and cloud-scale (<5–10 km) models are required to conduct studies of extreme precipitation events on hourly/daily time scales over limited spatial/time domains. Cloud-scale GCM and coupled GCMs are certainly desirable for better simulations of MCS over the global domain. However, such GCM simulations are highly labor-intensive and expensive for climate-scale long-term integrations. That is why most long-term GCM climate experiments, such as in CMIP6, are still expected to run at moderate-to-low resolution (>50–100 km) in the foreseeable future. Here, we show important results indicating that improved cumulus parameterization in a state-of-the-art GCM with moderate resolution can show MCS-like organization features for extreme tropical precipitation, on monthly time scales. Such an approach allows for the physics of extreme precipitation, such as MCS-like organization, to be explored and evaluated by precipitation and cloud observations on a global climatic scale, bridging the gap between meso-scale and low-resolution climate models.

**Supplementary Materials:** The following supporting information can be downloaded at: https://www.mdpi.com/article/10.3390/atmos14060995/s1, Figure S1. Spatial distribution of SST, Frequency of Occurrence (FOC), and the fraction of stratiform (large-scale) rain for top 5% monthly precipiptation; Figure S2. Spatial distribution of SST, precipitation efficiency (PE), outgoing longwave radiation (OLR) for top 5% monthly precipitation; Table S1. Extreme tropical monthly precipitation intensity threshold (mm/day) as a function of top-percentile rain rate for the entire tropics (30° S–30° N).

**Author Contributions:** Conceptualization, W.K.M.L. and L.R.L.; methodology, W.K.M.L., B.H. and K.-M.K.; software, B.H. and K.-M.K.; validation, W.K.M.L., B.H. and K.-M.K.; formal analysis, K.-M.K.; investigation, W.K.M.L. and K.-M.K.; resources, L.R.L., B.H. and K.-M.K.; data curation, B.H. and K.-M.K.; writing—original draft preparation, W.K.M.L.; writing—review and editing, L.R.L., B.H. and K.-M.K.; visualization, K.-M.K. and W.K.M.L.; supervision, W.K.M.L.; funding acquisition, W.K.M.L. and L.R.L. All authors have read and agreed to the published version of the manuscript.

**Funding:** This research was funded by the U.S. Department of Energy (DOE), the Office of Science, Biological, and Environmental Research as part of the Regional and Global Model Analysis Program Area under Grant Award #300426-00001, awarded to the University of Maryland from the Pacific Northwest National Laboratory (PNNL). The PNNL is operated for the DOE by the Battelle Memorial Institute under contract DE-AC05-76RL01830. Partial support was also provided by the NASA Modeling and Analysis Program, Federal Award Identification # 80NSSC21K1800, awarded to the University of Maryland and to the NASA Goddard Space Flight Center.

**Institutional Review Board Statement:** Not applicable.

**Informed Consent Statement:** Not applicable.

**Data Availability Statement:** The data used in this research were based on model outputs of the DOE/PNNL, available at https://portal.nersc.gov/archive/home/b/beharrop/www/e3sm_v1_pd_and_warming_experiments/e3sm_v1_control_simulations.tar (assessed on 1 June 2023).

**Conflicts of Interest:** The authors declare no conflict of interest.

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
