# Peer review of "Changing Characteristics of Tropical Extreme Precipitation–Cloud Regimes in Warmer Climates"

_atmosphere, doi:10.3390/atmos14060995_

Round 1

Reviewer 1 Report

Report on manuscript atmosphere-2398906

Changing characteristics of tropical extreme precipitation-cloud regimes in warmer climates

This manuscript used the Energy Exascale Earth System Model E3SM to study the characteristics of climate-scale tropical extreme precipitation in a warming climate. Authors used three ten-long years of simulation in the study, control, P4K, and 4xCO2. The paper is well presented and has the potential to be published, but I would like that the authors address the following issues:

Main Comments:

-        Keywords are missing.

-        Introduction:

o   The background behind the study is not really clear/presented, please extend your literature introduction.

o   authors need to motivate their selection of the used model(s), and highlight how their study will add to the literature. This is late mentioned in the model description but basically should come earlier in the introduction.

-        Figure 1 has three sub-figures; the caption need revision to include the left part.

-        L.175 – 176: please re-write

-        Punctuation symbols are missing at many equations.

-        Be consistent in using Fig. and Figure, and use \label and \ref{fig…} if you are using latex to avoid spacing problems. I

-        Be consistent in using climate-scale and climate scale.

-        Titles of axis are also missing in most figures 2,3,4, 5.

-        Re-unit the caption and the plots in Fig. 5 to the same page

-        Figure 6: what is (10-6kg/kg) in the caption?

-        Please extend your discussion of the statistical analysis.

-        per day and day-1 are the same, check!

-        The resolution of most figures is not that good.

-        The increase in the presented Cp is higher for land than ocean, please comment!

-        Authors need to pay attention for the references as well to follow the style of the journal, sometimes doi is given, sometimes not, please check!

Minor editing of English language required in few places, I highlighted few in the report.

Author Response

Reply to Reviewer 1  

Changing characteristics of tropical extreme precipitation-cloud regimes in warmer climates

This manuscript used the Energy Exascale Earth System Model E3SM to study the characteristics of climate-scale tropical extreme precipitation in a warming climate. Authors used three ten-long years of simulation in the study, control, P4K, and 4xCO2. The paper is well presented and has the potential to be published, but I would like that the authors address the following issues:

Main Comments:

-        Keywords are missing.

The following keywords have been added in the revised manuscript

-       Key words:  climate-scale extreme tropical precipitation; stratiform and convective precipitation; precipitation efficiency; meso-scale convective complex, surface warming vs. moistening; convective inhibition over land

- Introduction

o   The background behind the study is not really clear/presented, please extend your literature introduction.

We have added more detailed discussion of literature on extreme tropical precipitation.

Previous observational and climate modeling studies have shown that under global warming, the rate of increase in top 0.1% of tropical daily precipitation has been estimated to be near 10% K-1, significantly higher than those in the extratropics, which is limited by the thermodynamic rate of 6-7% K-1, governed by the Clausius Clapeyron relationship for atmospheric saturated moisture and temperature (2-5). Models and observations also showed that as the earth surface and the atmosphere warm up under anthropogenic CO2 radiative forcing, convection becomes more vigorous, and clouds grow faster, wider, and taller, producing more extreme precipitation (48).  An increasing number of recent studies [5, 49-51] have shown that extreme precipitation events attributable to GHG warming tend to occur preferentially in tropical/subtropical land and ocean with strong, and sustained organization of deep convection, embedded in extended areas of high anvil clouds associated with long-lived strong mesoscale convective systems (MCS). Even though such long-lived MCS occurs in less than 5% of tropical precipitation events in preferred climatological wet regions, they account for more than 40-50 % of the total extreme precipitation amount [52].  This could mean that extreme precipitation, which occurs on hourly/daily time scales, could have climatic signals in organization on monthly and longer time scales over specific land or oceanic regions, and even over the entire tropics. 

In spite of increasing reports of devastating destructive impacts on populated land regions, the scientific questions of whether extreme cloud-precipitation organization are a) fundamentally different, b) more or less intense, and/or frequent over land vs. ocean on climatic time scales remain uncertain.  In this paper, we will focus on addressing these questions and scientific rationales underlying them based on general circulation model (GCM) simulations.  However, because of GCM’s coarse resolution (>50-100km), MCS are not explicitly resolved, and not well simulated in traditional GCM cloud-precipitation parameterization.  More recently, MCS-like features have been simulated, and tracked in moderate resolution (50 km) GCM with improved physical cloud-precipitation parameterization that includes organization features that occur across scales (53, Dong et al. 2021).  In this study, we use the Department of Energy, Exascale Energy Earth System (E3SM) model which includes improved unified parameterization of clouds and precipitation types, to examine its capability in simulating MCS-like features, contributing to extreme tropical precipitation on climatic time scales. See further discussions on E3SM model physics in Section 2. 

  • authors need to motivate their selection of the used model(s), and highlight how their study will add to the literature. This is late mentioned in the model description but basically should come earlier in the introduction.

See reply to previous comments

-        Figure 1 has three sub-figures; the caption need revision to include the left part.

The figure labels a) and b) were somehow left out in the MDPI editing process.  Figure 1 has been revised with two main panels, illustrating a) cloud-precipitation processes in MCS, and b) condensation heating profiles separately for stratiform and convective rain (left subpanel) and for different fractions of stratiform vs. convective rain (right subpanel)

-        L.175 – 176: please re-write.  

The sentences have been rewritten to improve clarity.

       … To disentangle the effects of surface warming vs. CO2 radiative forcing respectively, the equilibrium solutions based on two separate AMIP runs identical to the Control have been conducted.  First, SST is increased by including an idealized plus-4K (P4K) anomaly, uniformly across the globe.  Second, SST anomaly (SSTA) and CO2 radiative forcing are imposed based on climatology of the last 30-year simulations of an abrupt 4 times CO2 (4xCO2) experiment with the coupled ocean-atmosphere version of the E3SM, as part of the Coupled Model Intercomparison Project phase-6 (CMIP6) [71]. 

-        Punctuation symbols are missing at many equations.

Equations are punctuated and numbered

-        Be consistent in using Fig. and Figure, and use \label and \ref{fig…} if you are using latex to avoid spacing problems.

“Figure “is fully spelled out in the revised paper.

-        Be consistent in using climate-scale and climate scale.

      Done

-        Titles of axis are also missing in most figures 2,3,4, 5.

Titles of axis have been included.

-        Re-unit the caption and the plots in Fig. 5 to the same page

      Done

-        Figure 6: what is (10-6kg/kg) in the caption?

       “Ice mass per Kg of air mass” is included in Fig. caption.

-        Please extend your discussion of the statistical analysis.

       We have included the following statements in the main text, and highlighted regions exceeding 95% statistical significance in Fig. 6, 7 and 8. 

       Next, we explore the capability of E3SM in simulating MCS-like extreme precipitation organization, with regard to increased contribution of stratiform (anvil) rain and enhanced PE in the production by freezing, and removal of cloud ice by melting and precipitation fallout.  Specifically, we have computed composite change patterns of P4K, 4XCO2 relative to the control, and 4xCO2 relative to P4K respectively in vertical profiles of key MCS quantities, i.e., cloud ice concentration, condensation heating, and large-scale vertical velocity (See Fig. 1a, b) as a function of precipitation intensity for the entire tropics, separately for land and ocean. 

-        per day and day-1 are the same, check!

      Change per day to day-1.

-        The resolution of most figures is not that good.

       All figures have been converted to EPS to improve clarity

-        The increase in the presented Cp is higher for land than ocean, please comment!

       CpT is higher over land than ocean, because of the much smaller heat capacity of land compared to ocean.  As a result, land temperature rises faster and higher than ocean for the same amount of heat input. In addition, the lack of land moisture sources results in less evaporative cooling.  Hence, under global warming, land temperature has to rise much higher compared to the ocean to increase outgoing longwave radiative cooling, needed to balance the land heating from CO2 greenhouse effect, and increased downward solar radiation from reduced clouds due to drying..

-        Authors need to pay attention for the references as well to follow the style of the journal, sometimes doi is given, sometimes not, please check!

      doi is included in all references

Comments on the Quality of English Language

Minor editing of English language required in few places, I highlighted few in the report.

Submission Date

28 April 2023

Date of this review

05 May 2023 16:29:18

Reviewer 2 Report

The authors demonstrate changes in tropical extreme precipitation and cloud regimes under different climate scenarios.

The manuscript is generally well written and deserves publication.

However, given the background and facilities of the authors and interest of the reader, it would have been very easy to redo/add 2 types of more realistic simulations (a) with higher horizontal resolution (ie 10-20 km which would allow you to talk more robustly about mesoscale convective systems) and (b) redo the runs in coupled mode. I know it is not the policy of MDPI to give authors time for revision etc. but this would give more weight.

While, as I said this is certainly one of the top 25% of the MDPI atmosphere manuscripts, the only thing I didn't find convincing is when you mention throughout the manuscript that you equal the increase in (stratiform) rainfall with an  increase in MCS. Your resolution doesnt allow this, neitehr is there a tracking, pdf or scale analysis of these systems, so it could just be that you have more saturated grid-cells (in contrast to "organized" convection).  I think if you cant demonstrate MCS you should tone down this a bit in the manuscript- In Figures 7,8 it is also apparent that the copnvection/troposphere just goes overall deeper as likely goes as you say the Hadley/Walker cells.
Finally, concerning Figure 9, while CIN and cAPE are useful, making the analysis of the terms near the surface is not useful=obvious result over oceans/land. To reflect the change in stability/circulation these terms should be analysed at 850 or 700 hPa

Author Response

Response to Reviewer 2  (revised text in blue)

Comments and Suggestions for Authors

The authors demonstrate changes in tropical extreme precipitation and cloud regimes under different climate scenarios.

The manuscript is generally well written and deserves publication.

However, given the background and facilities of the authors and interest of the reader, it would have been very easy to redo/add 2 types of more realistic simulations (a) with higher horizontal resolution (ie 10-20 km which would allow you to talk more robustly about mesoscale convective systems) and (b) redo the runs in coupled mode. I know it is not the policy of MDPI to give authors time for revision etc. but this would give more weight.

Thank you for your suggestion.  To clarify, we added the following discussion in the conclusion. Finally, we note that most high-resolution MCS-resolving meso-scale (10-20 km) and cloud-scale (<5-10 km) models are devoted to case studies of extreme events on hourly/daily time scale over limited spatial/time domains.   Simulations using high-resolution GCM and coupled GCMs are certainly desirable for better simulations of MCS.  However, they are highly labor intensive, and expensive for climate scale long-term integrations.  That is why most long-term GCMs climate experiments such as in CMIP6 are still expected to run at moderate-to-low resolution (>50-100 km) in the foreseeable future.  Here, we show the important results that improved cumulus parameterization in state-of-the-art GCM with moderate resolution can show “MCS-like” organization features for extreme precipitation.  Such an approach allows the physics of extreme precipitation such as MCS-like organization be explored, and evaluated by precipitation and cloud observation on a global climatic scale, bridging the gap between meso-scale and low-resolution climate models.    

While, as I said this is certainly one of the top 25% of the MDPI atmosphere manuscripts, the only thing I didn't find convincing is when you mention throughout the manuscript that you equal the increase in (stratiform) rainfall with an  increase in MCS. Your resolution doesnt allow this, neitehr is there a tracking, pdf or scale analysis of these systems, so it could just be that you have more saturated grid-cells (in contrast to "organized" convection).  I think if you cant demonstrate MCS you should tone down this a bit in the manuscript- In Figures 7,8 it is also apparent that the copnvection/troposphere just goes overall deeper as likely goes as you say the Hadley/Walker cells.
Finally, concerning Figure 9, while CIN and cAPE are useful, making the analysis of the terms near the surface is not useful=obvious result over oceans/land. To reflect the change in stability/circulation these terms should be analysed at 850 or 700 hPa

Thanks for your comments.  In the revised paper, we have taken care to tone-down, and emphasized that MCS scales are not resolved in GCM.  What we see are MCS-like organization features simulated by improved cumulus parameterization in the medium resolution (100km) E3SM.  At the time of revision, we became aware of one recent published paper (Dong et al. 2021) using a moderate resolution (50 km) GFDL GCM to detect MCS via a tracking algorithm for MCS based on threshold of brightness temperate, and areal coverage.  We have included reference to that work in the revised paper.  In the revision, we have emphasized what we saw are MCS-like organization based on improved parameterization in E3SM.

We have computed the CIN analysis at 850 and 700 hPa as suggested.   The anomalous RH /CIN at 850 and 700 hPa are much diluted over land, and merges with those over the ocean, reflecting the effect of LS downward motion and drying effect over land, as a result of the changes in the Walker/Hadley circulations. We already refer to the importance of the LS circulation in the discussion of enhanced LS downward motion over land in Fig.8.  As indicated in the concluding discussion, change in LS circulation, particularly the Walker circulation under global warming in affecting extreme precipitation organization over land vs. ocean are being investigated in CMIP6 GCMs and validation with precipitation observations in an ongoing investigation by our team.

Submission Date

28 April 2023

Date of this review
